# On the use of linguistic similarities to improve Neural Machine Translation for African Languages

**Pascal Tikeng**[*]
Department of Computer Science
NASEY
pascalnotsawo@gmail.com

**Brice Nanda**
Department of Computer Science
NASEY
bricenanda21@gmail.com

**James Assiene** [†]
assiene.james@gmail.com

## Abstract

In recent years, there has been a resurgence in research on empirical methods for machine translation. Most of this research has been focused on high-resource, European languages. Despite the fact that around 30% of all languages spoken worldwide are African, the latter have been heavily under investigated and this, partly due to the lack of public parallel corpora online. Furthermore, despite their large number (more than 2,000) and the similarities between them, there is currently no publicly available study on how to use this multilingualism (and associated similarities) to improve machine translation systems performance on African languages. So as to address these issues, we propose a new dataset (from a source that allows us to use and release) for African languages that provides parallel data for vernaculars not present in commonly used dataset like JW300. To exploit multilingualism, we first use a historical approach based on migrations of population to identify similar vernaculars. We also propose a new metric to automatically evaluate similarities between languages. This new metric does not require word level parallelism like traditional methods but only paragraph level parallelism. We then show that performing Masked Language Modelling and Translation Language Modeling in addition to multi-task learning on a cluster of similar languages leads to a strong boost of performance in translating individual pairs inside this cluster. In particular, we record an improvement of 29 BLEU on the pair Bafia-Ewondo using our approaches compared to previous work methods that did not exploit multilingualism in any way. Finally, we release the dataset and code of this work to ensure reproducibility and accelerate research in this domain.

## 1 Introduction

Machine Translation (MT) of African languages is a challenging problem because of multiple reasons. As pointed out by Martinus and Abbott (2019), the main ones are:

- Morphological complexity and diversity of African languages : Africa is home to around 2144 languages out of nowadays 7111 living languages (they thus make 30.15% of all living languages) with often different alphabets (Eberhard et al., 2020).

- Lack/Absence of large parallel datasets for most language pairs.

---

[*]NASEY : National Advanced School of Engineering Yaounde, Cameroon
[†]MILA-Quebec Artificial Intelligence Institute

- Discoverability: The existing resources for African languages are often hard to find.
- Reproducibility: Data and code of existing research are rarely shared, making it difficult for other researchers to reproduce the results properly.
- Lack of benchmarks: Due to the low discoverability and the lack of research in the field, there are no publicly available benchmark or leaderboard to compare new machine translation techniques to.

However, despite the strong multilingualism of the continent, previous research focused solely on translating individual pairs of language without taking advantage of the potential similarities between all of them. The main purpose of this work is to exploit this multilingualism, starting from a few languages spoken in Central and West Africa, to produce better machine translation systems.

Our contributions can be summarised as follows :

1. We provide new parallel corpora extracted from the Bible for several pairs of African vernacular [3] that were not available until now as well as the code used to perform this extraction.
2. We present a method for aggregating languages together based on their historical origins, their morphologies, their geographical and cultural distributions etc... We also propose of a new metric to evaluate similarity between languages : this metric, based on language models, doesn't require word level parallelism (contrary to traditional like (Levenshtein, 1965) and (Swadesh, 1952)) but only paragraph level parallelism. It also takes into account the lack of translation of words present in (Swadesh, 1952) but not in African vernaculars (like "snow").
3. Using the language clusters created using the previous similarities, we show that Translation Language Modelling (Lample and Conneau, 2019) and multi-task learning generally improve the performance on individual pairs inside these clusters.

Our code, data and benchmark are publicly available at `https://github.com/Tikquuss/meta_XLM`

The rest of the paper is organised as follows: In section 2, we discuss the motivation of this work i.e. problems faced by African communities that could be solved with machine translation systems. Related work is described in 3 and relevant background is outlined in 4. In section 5, we describe our dataset and provide more details on the language aggregation methods. In section 6 we present the experimental protocol and the results of our methodology on machine translation of 3 African (Cameroonian) vernaculars. Lastly, we summarise our work and conclude in 7.

## 2 Motivation

Africa, because of its multilingual nature, faces a lot of communication problems. In particular, most African countries have colonial languages (French, English, Portuguese, etc...) as official languages (with sometimes one or two African dialects). The latter are those taught in schools and used in administrations and workplaces (which are often located in urban areas). In contrast, rural and remote areas citizens mainly speak African dialects (especially the older ones and the less educated ones). This situation creates the following fairness and ethical concerns (amongst others):

- Remote area population often have difficulties communicating during medical consultations since they mainly speak the local languages, contrary to the doctors who mostly speak French, English, etc.... Similar situations arise when NGOs try to intervene in rural regions.
- This discrepancy between spoken languages across different regions of individual African countries make the spread of misinformation easier (especially in rural regions and especially during election periods).
- Young Africans (in particular those living in urban areas) struggle to learn their vernaculars. Lack of documentation and schools dealing with African languages as well as the scarcity of translators and trainers make the situation even more complicated.

For all of the reasons above, it is essential to set up translation systems for these languages.

---

[3] `https://en.wikipedia.org/wiki/Vernacular`

# 3 Related work

Several scholarships have been conducted on machine translation for African languages, Wilken et al. (2012) performed statistical phrase-based translation for English to Setswana translation. This research used linguistically-motivated pre and post processing of the corpus in order to improve the translations. The system was trained on the Autshumato dataset (Groenewald and Fourie, 2009) and also used an additional monolingual dataset. They reached a BLEU score of 28.8. McKellar (2014) used statistical machine translation for English to Xitsonga translation. The models were trained on the Autshumato data (Groenewald and Fourie, 2009), as well as a large monolingual corpus. A factored machine translation system was used, making use of a combination of lemmas and part of speech tags. It obtained a BLEU score of 37.31. Niekerk (2014) used unsupervised word segmentation with phrase-based statistical machine translation models. These models translate from English to Afrikaans, N. Sotho, Xitsonga and isiZulu. The parallel corpora were created by crawling online sources and official government data and aligning these sentences using the HunAlign software package. Large monolingual datasets were also used. They reached a BLEU score of 71 on English to Afrikaans and 37.9 on English to N.Sotho. Wolff and Kotze (2014) performed word translation for English to isiZulu. The translation system was trained on a combination of Autshumato, Bible, and data obtained from the South African Constitution. All of the isiZulu text was syllabified prior to the training of the word translation system.

Abbott and Martinus (2018) trained Transformer models for English to Setswana on the parallel Autshumato dataset (Groenewald and Fourie, 2009) and obtained a BLEU score of 33.53. Data was not cleaned nor was any additional data used. This is the rare study reviewed that released datasets and code. Martinus and Abbott (2019) used Google Translate to perform English to Afrikaans translation and achieved a BLEU score of 41.18 To the best of our knowledge, Masakhane (2020) are the only ones to have carried out research on a wide variety of African languages. They used the JW300 dataset (Agić and Vulić, 2019) from opus collection [4] and trained Transformer models on it. However, they did not do any preliminary linguistic study of their languages of interest. Lastly, Emezue and Dossou (2020) provided a benchmark for Fon to French translation using the same JW300 dataset (Agić and Vulić, 2019). They used a GRU (Chung et al., 2014) sequence to sequence model with attention (Bahdanau et al., 2015) and obtained a BLEU score of 30.55. These last two studies are ones of the very few to have released datasets and code.

Regarding the use of multilingualism in machine translation, (Johnson et al., 2016) applied multi-task learning to a single encoder-decoder LSTM model to translate between English, French and German. The best gain of performance of their many-to-many model was 0.43 BLEU on *(French,English)* compared to a single-task model trained on that pair. However, its worse decrease of performance was 2.75 BLEU on *(English, German)*

Ha et al. (2016) also worked on English, French and German. Their best gain of performance compared to a single task model was 2.64 BLEU on *(English, German)* in an under-resourced scenario. They used an original approach called *mix-source* where the pair *(German, German)* was added to the translation model training data. The goal was to help the model obtaining a better representation of the source side and learn the translation relationships better. They argued that including monolingual data in this way might also improve the translation of some rare word types such as named entities.

# 4 Background

## 4.1 Neural Machine Translation (NMT)

Machine Translation (MT) refers to fully automated software that can translate source content into target languages. Neural Machine Translation (NMT) is an approach to MT that uses an artificial neural network to predict the likelihood of a sequence of words, typically modeling entire sentences in a single integrated model. NMT aims to translate an input sequence from a source language to a target language. An NMT model usually consists of an encoder to map an input sequence to hidden representations, and a decoder to decode hidden representations to generate a sentence in the target language.

---

[4] http://opus.nlpl.eu/JW300.php

NMT has made rapid progress in recent years. The attention module is first introduced by Bahdanau et al. (2015), which is used to better align source words and target words. The encoder and decoder can be specialized as LSTM (Hochreiter and Schmidhuber, 1997; Sutskever et al., 2014; Wu et al., 2016), CNN (Gehring et al., 2017) and Transformer (Vaswani et al., 2017). A Transformer layer consists of three sublayers, a self-attention layer that processes sequential data taking the context of each timestep into consideration, an optional encoder-decoder attention layer that bridges the input sequence and target sequence which exists in decoder only, and a feed-forward layer for non-linear transformation. Transformer achieves the state-of-the-art results for NMT (Barrault et al., 2019).

## 4.2   Cross-lingual Language Model (XLM) Pretraining

Recently, pre-training techniques, like ELMo (Peters et al., 2018), GPT/GPT-2 (Radford et al., 2018), BERT (Devlin et al., 2019), cross-lingual language model (briefly, XLM) (Lample and Conneau, 2019), XLNet (Yang et al., 2019) and RoBERTa (Liu et al., 2019) have attracted more and more attention in machine learning and natural language processing communities. The models are first pre-trained on large amount of unlabeled data to capture rich representations of the input, and then applied to the downstream tasks by either providing context-aware embeddings of an input sequence (Peters et al., 2018), or initializing the parameters of the downstream model (Devlin et al., 2019) for fine-tuning. Such pre-training approaches lead to significant improvements on natural language understanding tasks.

A Cross-lingual language model (Lample and Conneau, 2019) is used to obtain a better initialization of sentence encoders for zero-shot cross-lingual classification, a better initialization of supervised and unsupervised neural machine translation systems, a language models for low-resource languages and unsupervised cross-lingual word embeddings. This is done through four objectives:

- **Shared sub-word vocabulary** : XLM processes all languages with the same shared vocabulary created through Byte Pair Encoding (BPE) (Sennrich et al., 2015)

- **Causal Language Modeling (CLM)** : XLM causal language modeling (CLM) task consists of a Transformer language model trained to model the probability of a word given the previous words in a sentence $P(\omega_t|\omega_1,...,\omega_{t-1},\theta)$.

- **Masked Language Modeling (MLM)** : XLM consider the masked language modeling (MLM) objective of Devlin et al. (2019), also known as the Cloze task (Taylor, 1953). Following Devlin et al. (2019), XLM sample randomly 15% of the BPE tokens from the text streams, replace them by a [MASK] token 80% of the time, by a random token 10% of the time, and keep them unchanged 10% of the time. Differences between XLM approach and the MLM of Devlin et al. (2019) include the use of text streams of an arbitrary number of sentences (truncated at 256 tokens) instead of pairs of sentences.

- **Translation Language Modeling (TLM)** : Both the CLM and MLM objectives are unsupervised, only require monolingual data, and cannot be used to leverage parallel data when it is available. XLM introduce a new translation language modeling (TLM) objective for improving cross-lingual pretraining. TLM objective is an extension of MLM, where instead of considering monolingual text streams, parallel sentences are concatenated. XLM randomly mask words in both the source and target sentences. To predict a word masked in an source sentence, the model can either attend to surrounding source words or to the target translation, encouraging the model to align the source and target representations. In particular, the model can leverage the target context if the source one is not sufficient to infer the masked source words. XLM also reset the positions of target sentences to facilitate the alignment.

In this work, we used the XLM [5] framework to train and evaluate our models.

---

[5] https://github.com/facebookresearch/XLM

# 5 Methodology

## 5.1 Data

YouVersion provides biblical content in 1,756 languages [6] among which many (local) African ones. We have extracted this content in the desired languages to build our dataset. In particular, for some countries like Cameroon, YouVersion provide more languages (22 for the latter) than JW300 (Agić and Vulić, 2019) (18 for Cameroon) that is the most used data source for work on African languages (e.g Masakhane (2020)). We release our adaptable extraction script along with this publication. The number of sentences per language is given in figure 1.

| English | French | Bafia | MASSANA | Bulu | Dii | Doyayo | Dun |
|---------|--------|-------|---------|------|-----|--------|-----|
| 30901 | 31296 | 7950 | 31128 | 28895 | 28895 | 7900 | 7916 |
| Ejagham | Fulfulde | Ghomala | Guidar | Guiziga | Gbaya | Kapsiki | Limbum |
| 7890 | 30840 | 7942 | 7915 | 31084 | 31092 | 31095 | 7919 |
| Ewondo | Mofa | Ngiemboon | Mofu | Peere | Samba | Tupurri | Vute |
| 7944 | 7945 | 7929 | 7941 | 7905 | 7905 | 31268 | 7909 |

Figure 1: Number of sentences per language. The number of sentences for a language pair $(L_1, L_2)$ is equal to $min(|L_1|, |L_2|)$ where $|L_i|$ represents the number of sentences of the language $L_i$

The Bible is one of the rare publicly available online corpora for low-resource languages such as those spoken in Africa. Its structure (made of books, chapters and verse numbers) allows to automatically and unambiguously identify parallel paragraphs (verses) across all languages. However, it suffers from several flaws for machine translation:

1. In general, YouVersion only provides the New Testament for African languages, consisting of about 7958 verses only. In the case of languages for which both New and Old Testament are available, one can retrieve around 31170 verses.

2. The Bible contains a lot of redundant sentences ("Truly, I say to you/I tell you" for example), verses made up of characters and places names etc... Making the translation task easier than if a closer to real world dataset like WMT was used. Its antiquated writing style is also far from contemporary style.

3. The alignment information is limited to verses rather than sentences unlike the JRC-Acquis corpus (Steinberger et al., 2006) for instance.

Christodouloupoulos and Steedman (2015) gives more details on pros and cons of using the Bible as a machine translation corpus.

Following Lample and Conneau (2019), we process all the languages with the same shared vocabulary created using Byte Pair Encoding (BPE) (Sennrich et al., 2015). Due to the lack of specific tokenizers for African languages, we used an english one on all of them (we performed tokenization before applying the BPE algorithm).

## 5.2 Languages similarities and aggregations

### 5.2.1 Historical approach for languages aggregation

From a historical point of view, just like African borders, African languages have undergone the movements of the early inhabitants of the continent. These migrations have shaped and modified the languages spoken in multiple regions and led to resemblances between some of them. For these reasons, we examined and used languages aggregations proposed by Eberhard et al. (2020) that are based on their origins and evolutions.

The intersection between our dataset and Eberhard et al. (2020) analysis led to the following languages clusters:

---

[6] `https://www.bible.com/en/languages` (October 22, 2021)

- Cluster 1 : (**Bafia, Ewondo, Bulu**). These languages are all part of the Niger-Congo, **Northwest**[7] Narrow Bantu family and spoken in the central and southern regions of Cameroon.

- Cluster 2 : (**Ngiemboon, Ghomala, Limbum**). They are part of the Niger-Congo, **Mbam-Nkam**[8] family and spoken in the West Cameroon.

- Cluster 3 : (**Dii, Doyago, Samba Leko, Peere**). These languages are spoken in the Northern region of Cameroon and belong to the Niger-Congo, **Adamawa**[9] family

- Cluster 4 : (**Guiziga, Mofa, Mofu-Gudur, Kapsiki, Guidar**). They are also spoken in Northern Cameroon but have been classfied as Afro-Asiatic languages. In particular, they belong to the **Biu-Mandara**[10] family.

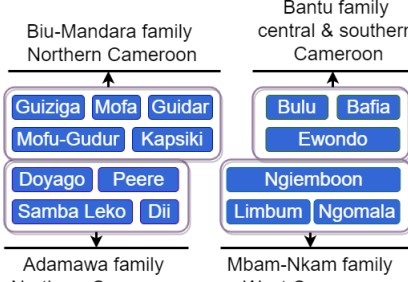

Figure 2: Historical approach for languages aggregation

In the rest of this work we used the first cluster mentioned above for the experiments : it is the smallest of the 4 and is made of some of the most spoken languages in Cameroon (compared to the others listed above)

### 5.2.2 Language model based approach for lingustic similarities

Despite providing a historically grounded methodology to identify languages families, methods such as Eberhard et al. (2020) are time-consuming and require a lot of background knowledge on specific population migration throughout history. Such in-depth knowledge might not be available for all languages. Besides, they don't provide a quantitative measure for languages similarity, making it difficult for one to assess if a language $L_i$ is closer to a language $L_j$ than to a language $L_k$. Quantitative metrics are necessary for well known clustering algorithms like K-Means or Hierarchical Clustering.

Multiple linguistic similarity metrics have been proposed in the past. The most popular are the ones used in lexicostatistics among which Swadesh lists (Swadesh, 1952) and Levenshtein distance (Levenshtein, 1965). Nonetheless, the latter require data parallelism at word level and dictionaries might not be freely available online for all languages pairs. Additionally, Swadesh lists (Swadesh, 1952) are based on a small number of words that might not even exist in all languages. For example, the word "snow" (present in the Swadesh 207 list) doesn't have a translation in Bafia (likely because it never snows in this region).

To alleviate these problems, we propose a language model based metric for linguistic similarity that requires paragraph level parallelism only. This approach formalises (using language models) the following human intuition : *a child whose native language is $L_1$ may think that a language $L_2$ is similar to $L_1$ if words (possibly with the same meaning) in $L_2$ "sounds"/"looks" like words in $L_1$.*

For example, the english word "bilingual" is translated in french by "bilingue" and in spanish by "bilingüe" but its polish translation is "dwujezyczny". A french speaker can easily think that the spanish "bilingüe" is a french word both at the spelling level because of common syllables ("bi", "lin" and "gu"/"gü") and at the pronunciation level. Nevertheless, it is unlikely for the polish "dwujezyczny" to be seen as a french word. Following the success of recent language models like GPT-2 Radford et al. (2019), the previous french speaker can be seen as a very powerful french language model $lm_{fr}$. If we evaluate it on both "bilingüe" and "dwujezyczny", it should give a higher probabilty to the spanish "bilingüe" than to the polish "dwujezyczny" because of the french "bilingue". More formally, we expect the BPE encoding to capture syllables like "bi","lin" and "gu"/"gü"[11] so that if $P_{lm_{fr}}(bilingue) = P_{lm_{fr}}(bi) * P_{lm_{fr}}(lin|bi) * P_{lm_{fr}}(gu|bi, lin) * P_{lm_{fr}}(e|bi, lin, gu)$ is relatively high then $P_{lm_{fr}}(bilingüe) = P_{lm_{fr}}(bi) * P_{lm_{fr}}(lin|bi) * P_{lm_{fr}}(gü|bi, lin) * P_{lm_{fr}}(e|bi, lin, gü)$ should be greater than $P_{lm_{fr}}(dwujezyczny)$.

---

[7] https://www.ethnologue.com/subgroups/a-4

[8] https://www.ethnologue.com/subgroups/mbam-nkam

[9] https://www.ethnologue.com/subgroups/adamawa

[10] https://www.ethnologue.com/subgroups/biu-mandara

[11] "ü" can become "u" during lemmatization

At a paragraph level, we expect that given two parallel paragraphs $C_{spanish}$ and $C_{polish}$, $lm_{fr}$ gives a higher probability to $C_{spanish}$ i.e. $P_{lm_{fr}}(C_{spanish}) > P_{lm_{fr}}(C_{polish})$

Given a trained language model $LM_0$ in language $L_0$ and two parallel corpora $C_0$ in $L_0$ and $C_1$ in language $L1$, we define the LM-similarity between $L_0$ and $L_1$ according to $LM_0$ and $C_0$ by

$$LMS_{LM_0,C_0}(L_0, L_1) = \frac{PPL_{LM_0}(C_1) - PPL_{LM_0}(C_0)}{PPL_{LM_0}(C_0)}$$

Where $PPL_{LM_0}(C_i)$ is the perplexity of the language model $LM_0$ on the corpus $C_i$. It is important to precise that Accuracy (percentage of tokens correctly predicted by the language model) can be used in lieu of Perplexity (this may lead to different similarities between languages).

Despite the fact that this metric is not symmetric, it can be used to build clusters using a hierarchical clustering approach. We start by a set of 2 languages $G_1 = \{L_1, L_2\}$, the language $L^*$ that will be added to $G_1$ can be determined using the following formula (inspired by the unweighted average linkage clustering criteria) :

$$L^* = \arg\min_{L_i \in \mathcal{L} \setminus G_1} \frac{1}{2} \left( LMS_{LM_1,C_1}(L_1, L_i) + LMS_{LM_2,C_2}(L_2, L_i) \right)$$

.

In other words, $L^*$ is the language in the set of possible languages $\mathcal{L}$ (but we exclude $G_1 = \{L_1, L_2\}$) that minimises the average "distance" (LM-similarity) from $L_1$ and $L_2$. By recursively applying this formula, one can build a cluster of any size.

The proposed similarity metric should be interpreted more as a dissimilarity, hence the equation normalizing a perplexity difference according to a base perplexity, to allow it to be zero on the base language and the base corpus. The word "similarity" here is only an abuse of language, to signify the fact that obenuous languages are similar in the sense of this metric. This method is mainly motivated by the fact that the resources needed to apply other methods such as those mentioned above are not available. M. Eberhard et al. (2019) has been easy to replicate in that it directly provides groups of similar languages: we have just made an intersection between these groups and the languages in our dataset. On the other hand, a method like (Swadesh, 1952) requires parallelism of data at the word level, which we did not have and could not find: this is one of the limitations of this work..

## 6   Experiments and Results

For each ordered language pair in the cluster of interest *(Bafia, Ewondo, Bulu)*, we examine the machine translation performance (using BLEU score (Papineni et al., 2002)) on that pair if we add a third language to the pair fo form a cluster. The model will thus become a multi-task machine translation model capable of translating 3 languages rather than 2. The language (among the 22 in our dataset) to add is chosen in 3 ways in our work:

1. Using a **historical** approach like Eberhard et al. (2020). For the pair *(Bafia, Ewondo)* we will add *Bulu*. It corresponds to the column **Historical** in Table 2

2. Using the **LM-similarity** (as described in 5.2.2). For the pair *(Bulu, Ewondo)*, it will be *Mofa*. It corresponds to the column **LM** in Table 2

3. Purely at **random** : it will serve as a baseline to make sure that the difference in performance is due to the particular choices of our methodology rather than multi-task learning solely. It corresponds to the column **Random** Table 2

Table 1: Languages used respectively for the Random, Historical and LM columns of Table 2. The first column represents the reference pairs, the columns Random, Historical and LM contain respectively the languages that have been added to form the clusters that gave the results of the corresponding columns in Table 2.

| Pair | Random | Historical | LM |
|---|---|---|---|
| Bafia-Bulu | Du_ n | Ewondo | Mofa |
| Bafia-Ewondo | Guidar | Bulu | Bulu |
| Bulu-Ewondo | Dii | Bafia | Mofa |

Having identified our clusters and to further exploit multilingualism, we follow the approach of Lample and Conneau (2019) by pre-training the model's encoder with MLM and TLM objectives

(without CLM) on the cluster languages, then fine-tune the model on these same languages. The MLM pre-training was used as the languages did not have the same number of sentences : for example for the Bafia-Bulu pair, we have 7950 parallel sentences (representing the number of sentences of the Bafia language) but 28895 sentences for Bulu (i.e. 28895 - 7950 = 20945 non-parallel sentences). The TLM is naturally motivated by XLM : encourage the model to align source and target sentence representations (especially by taking advantage of the target context if the source context is not sufficient to infer hidden source words).

All our models are tagged models, and the tag for a given language is its id in the list of languages used. Indeed, in addition to positional and token embedding, XLM does language embedding. This is motivated by several works. Caswell et al. (2019) add a specific token (the <BT> tag) at the beginning of the synthetical sentences produced using back-translation to distinguish them from the available non-synthetical sentences. Johnson et al. (2017) do multilingual MT by adding a special token at the beginning of each sentence of the same language indicating the source language, and do not change anything about the architecture of classical MT models. Yamagishi et al. (2016) add a tag specific to the voice (passive, active) of the target sentence in the source sentence so that the sentences produced by the model keep the voice of the source sentence. Kuczmarski and Johnson (2018) do the same, but more for gender (male and female gender).

We use the average BLEU score on the validation sets across all the languages as stopping criterion. Due to the multi-task nature of the model, this criterion is preferred to validation BLEU score on the pair of interest : in some cases, the model can first have low, stagnating performance on that pair for a large number of epochs (greater than the patience, threshold to stop the training) but at the same time have good performance on the other pairs (leading to an increase of average BLEU score). At the end of a training with average BLEU score as stopping criterion, the pair of interest "catches up" with the other pairs and obtain high scores.

We also compare the performance with model trained: 1) On the pair of interest without any form of MLM/TLM pre-training (Column **None** in Table 2). 2) On the pair of interest with MLM+TLM pre-training (Column **Pair** in Table 2)

### 6.1 Training details

We used the default settings mentioned in the documentation of the XLM framework : Transformer architectures with 1024 hidden units, 8 heads, GELU activations (Hendrycks and Gimpel, 2016), Dropout rate of 0.1 and learned positional embeddings, Adam optimizer (Kingma and Ba., 2014), a linear warmup (Vaswani et al., 2017) and learning rates of 0.0001 for both XLM pre-training (MLM and TLM, without CLM) and machine translation. For the MLM and TLM pre-training, we used the averaged perplexity over all the languages as a measure of performance and stopping criterion for training, and for machine translation, we used the average BLEU score across all the languages pairs of the models. The patience was set to 20 for all models.
The models were trained on one 48GB NVIDIA Quadro RTX 8000 GPU for *Random*, *Historical* and *LM* and on a 11GB NVIDIA RTX 2080 Ti for *None* and *Pair*.
In all our experiments, we used 80% of the dataset as training data, 10% as validation data and 10% as test data.

### 6.2 Results, discussion and limitations

Table 2 describes the results of the baselines mentioned in 6 and our models. We notice that using our historical approach systematically leads to a strong boost of performance and always outperforms the *None*, *Pair* and *Random* baselines. As the aggregation approach that led to the best results, we evaluated its impact on translating African vernaculars to European languages. We chose french as it is the most spoken language in Cameroon resulting in the cluster *(Bafia, Bulu, Ewondo, French)* on which MLM + TLM pretraining and multi-task learning were done and compared to *None* and *Pair* baselines. Again, the historical approach outperformed all the baselines with notably a gain of 22 BLEU on the pair *French-Ewondo*.

Our LM-similarity approach systematically performs better than the *None* and *Pair* baselines, suggesting that adding a language using this metric and do multi-task learning on the resulting cluster is a good heuristic even if one only cares about a specific pair of languages. It outperforms the baseline *Random* for 4 out of the 6 pairs of languages. According to this similarity, the closest language to

the pair (*Bafia, Ewondo*) is *Bulu*, thus reconstituting the historical cluster identified by 5.2.1 without any human supervision. Randomly choosing this particular language would have happened with a $\frac{1}{20} = 5\%$ probability ($20 = 22 - 2$, there are 22 languages in our dataset and we exclude 2 languages, *Bafia* and *Ewondo*, since we want to add another language to this pair).

The fact that the *Random* baseline performs worse than models derived from our methodology 10 times out of 12 suggests that the performance boost is not solely due to the multi-task nature of the model but mostly because of the good choices of languages to add. It is worth noticing that despite the MLM and TLM pre-training, the *Random* baseline performs worse than the baseline *Pair* 4 times out of 6. This suggests that randomly adding languages may often hurt the performance compared to a simple model trained on the pair of interest.

Table 2: Machine Translation BLEU scores. The rows correspond to the pairs of interest on which BLEU scores are reported. The column None is a baseline : it represents the BLEU score of a model trained on the pair without any MLM or TLM pre-training. The column Pair is a baseline : it represents the BLEU score of a model trained on the pair with MLM and TLM pre-training. The column Random is also a baseline : it is the BLEU score of a 3 languages multi-task model where the language added was chosen purely at random. The column Historical refers to the BLEU score of our 3 languages multi-task model where the language added was chosen using clusters identified in 5.2.1. The column LM describes the BLEU score of our 3 languages, multi-task model where the language added was chosen using the LM similarity ad described in 5.2.2 (Ba = Bafia, Bu = Bulu, Ew = Ewondo, Fr = French)

| Pair | None | Pair | Rand | Hist | LM | Pair | None | Pair | Hist |
|------|------|------|------|------|------|------|------|------|------|
| Ba-Bu | 9.19 | 12.58 | 23.52 | **28.81** | 13.03 | Fr-Bu | 19.91 | 23.47 | **25.06** |
| Bu-Ba | 13.5 | 15.15 | 24.76 | **32.83** | 13.91 | Bu-Fr | 17.49 | 22.44 | **23.68** |
| Ba-Ew | 9.30 | 11.28 | 8.28 | **38.90** | 38.90 | Fr-Ba | 14.48 | 15.35 | **30.65** |
| Ew-Ba | 13.99 | 16.07 | 10.26 | **35.84** | 35.84 | Ba-Fr | 8.59 | 11.17 | **24.49** |
| Bu-Ew | 10.27 | 12.11 | 11.82 | **39.12** | 34.86 | Fr-Ew | 11.51 | 13.93 | **35.50** |
| Ew-Bu | 11.62 | 14.42 | 12.27 | **34.91** | 30.98 | Ew-Fr | 10.60 | 13.77 | **27.34** |

We believe that the difference between Random and Pair is not due to a disparity in data. First of all, they come from the same domain, the biblical domain. With regard to the size of the dataset :

- For Bafia-Bulu, the three languages added are Du_n, Ewondo and Mofa for Random (R), Historical (H) and LMS (Language Model-based approach for linguistic Similarity) respectively : this makes three clusters of almost the same size in terms of data, since only Bulu (present in all three clusters) among all these languages has 28895 sentences, the others have an average of 7939 (average of the sentences of the languages present only in the new testament).

- Same thing for the Bulu-Ewondo pair with Dii, Bafia and Mofa added as languages for R, H and LMS respectively.

- The pair which is an exception to this rule is Bafia-Ewondo, for which the languages added are respectively Guidar, Bulu and Bulu for R, H and LMS : H and LMS thus have an advantage over R with Bulu, but the gain is quite as good as for the other clusters.

This difference may come from the natural similarity in the Historical clusters. In Cameroon you will have difficulty, if you are not very good at speaking these languages, in differentiating between the languages of the same historical cluster (both alphabetically and orally). On the other hand, you can very easily, even if you are not familiar with these languages, make the difference between two languages of two different clusters.

We hypothesize that a multi-task machine translation model (potentially with MLM and TLM pre-training) is capable of exploiting similarities between *similar* languages. A similar phenomenon can be observed with human beings : it is easier for someone that speaks French and Spanish to learn Italian (since they are all Latin languages) rather than polish (that is a Lechitic language).

**Limitations** :

- Out Of Distribution (OOD) generalization impossible, as well as a practical use of our method, because of the use of data from the biblical domain
- Due to the lack of specific tokenizers for African languages, we used an english. The development of spacialized tokenizers like the one proposed by Dossou and Emezue (2021) is left as a perspective to this work.
- The method used in this work to create the language clusters is only one agglomerative method among many others. We did not find it necessary to use very elaborate methods since the cluster was of size 3 (the historical cluster 1 has three languages).

**Negative Social Impact :** The authors do not foresee negative social impacts of this work specifically. Our theoretical work aims at showing the advantage of combining similar languages for machine translation, and the practical implications of this work (subject to the development of a more appropriate dataset) are mainly aimed at solving the problems related to language barriers in Africa

## 7 Conclusion

In this work, we studied the impact of using multilingualism to improve machine translation of African languages. Our first contribution is the creation and public release of a new dataset (as well as the code used to create it) that increases the number African languages for which parallel machine translation corpora is available. This study focused on pairs derived from a cluster of 3 of the most spoken languages in Cameroon. Our second contribution is the proposal of a new metric to evaluate similarity between languages : this metric, based on language models, doesn't require word level parallelism (contrary to traditional like (Levenshtein, 1965) and (Swadesh, 1952)) but only paragraph level parallelism. It also takes into account the lack of translation of words present in (Swadesh, 1952) but not in African languages (like "snow"). Translation Language Modeling and multi-task learning were used to exploit multilingualism. Our third contribution is the set of empirical evidences that doing multi-task learning on multiple similar languages generally improves the performances on individual pairs. The choice of the similarity is crucial since in our work, 4 times out of 6 a random choice leads to a decrease of performance. On the other hand, using our historical approach (derived from Eberhard et al. (2020)) systematically led to a boost of performance on this dataset and our LM similarity approach improved them 4 times out of 6.

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

## A  Appendix

**Additional details regarding motivations**   In relation to point 2 of the motivation section (**this discrepancy between spoken languages across different regions of individual African countries make the spread of misinformation easier (especially in rural regions and especially during election periods**)), we take an example relating to the election period. During this period, the government sends people to explain to people in rural areas what the vote is going to be used for and how they should vote. But it is common for them to do this in a way that favours certain political parties instead of simply translating the law, for example. Let's imagine that the text says in English "... You must therefore choose one and only one candidate between Candidate A and Candidate B." (... vous devez donc choisir un et un seul candidat entre le candidat A et le candidat B) and the translator says in French (to favour party A): "... vous devez donc choisir le candidat A et non le candidat B" (You must therefore choose candidate A and not candidate B). On the other hand, the information that is broadcast on television or radio is in French or English. If the villagers had access to a tool allowing them to translate the legal texts (which are in French or English) into their mother tongue, this could be avoided.

**Someone may ask "Why learning a different unrelated language should hurt other languages I know?"**   In our case, language learning takes place simultaneously, and we believe that learning several languages that are not similar (in the sense of the two similarities mentioned above) simultaneously makes the learning task more difficult. We also believe that the gain in performance at the level of the historical clusters comes from a natural similarity in the Historical clusters. In Cameroon you will have difficulty, if you are not very good at speaking these languages, in differentiating between the languages of the same historical cluster (both alphabetically and orally). On the other hand, you can very easily, even if you are not familiar with these languages, make the difference between two languages of two different clusters

