# OpenReview forum: "On the use of linguistic similarities to improve Neural Machine Translation for African Languages"
_ICLR.cc/2021/Conference — Reject_

### Official Review · AnonReviewer2 · 2020-10-27

**Rating:** 3
**Confidence:** 4

**Review:**

This paper explores the problem of machine translation for African languages. The authors show that trilingual models outperform bilingual models for a set of languages spoken in Cameroon, and explore two strategies to find the best auxiliary language for a given translation pair: a manual one based on their linguistic relationship, and an automatic one based on language modelling.

I agree with the authors that African languages have been underrepresented in NLP research in general and machine translation in particular. In that regard, I think that their motivation is truly admirable, and I would like to encourage them to keep working on this direction.

Unfortunately, I think that the work itself does not meet the standards of the conference in its current form. While the focus on African languages is new and interesting, I find that the paper has little substance beyond that. According to the authors the paper makes 3 main contributions, which I feel are too narrow or otherwise questionable as discussed next:

1) "Our first contribution is the creation and public release of a new dataset." The dataset in question is a parallel version of the Bible. There have already been some efforts to extract parallel corpora from the Bible, which are not even discussed in the paper. For instance, Mayer & Cysouw (2014) [1] report covering 830 languages, which are likely to include the ones explored in the paper. Moreover, the authors extracted the corpus from an existing website with Bible versions in >1000 languages, so this was just a scraping effort. In connection to that, I am not sure if the authors considered potential copyright restrictions when releasing their corpus.

2) "Our second contribution is the proposal of a new metric to evaluate similarity between languages". The proposed method measures the similarity between two languages, L1 and L2, by applying a language model trained on L1 to L2. This looks like a rather simplistic approach and, in the lack of any other baseline, it is hard to get a sense of how good it is. Also, I am not familiar with this topic, but there is certainly some work on automatically identifying similar languages. For instance, [2] shows that the degree of overlap in the BPE vocabulary of different languages is already a good indicator of the their linguistic proximity.

3) "Our third contribution is the set of empirical evidences that doing multi-task learning on multiple similar languages generally improves the performances on individual pairs." There were already in-depth studies showing that multilingual training is helpful for low-resource machine translation, see for instance [3]. Unfortunately, this is not even cited in the paper.

In connection to the last point, the authors limit themselves to trilingual models and the main focus of their work is on identifying the most suitable auxiliary language. An obvious baseline that is missing in the paper is to train a multilingual model on the combination of all languages.

In addition, the models used by the author are usually trained at a large scale, and might not work well with very small datasets like the Bible. Even a phrase-based statistical machine translation system might work better in this setup, and I overall miss better baselines.

[1] http://www.lrec-conf.org/proceedings/lrec2014/pdf/220_Paper.pdf
[2] https://aiide.org/ojs/index.php/AAAI/article/view/4677
[3] https://arxiv.org/abs/1907.05019

---

### Official Review · AnonReviewer4 · 2020-10-28
**Small contribution on transfer in MT for underresearched African languages**

**Rating:** 5
**Confidence:** 4

**Review:**

This paper motivates clearly the need for research in machine translation of underresourced (and thus underresearched) African languages, and proposes ways to aid the training of MT systems using data from related languages. The main contribution is that two ways (and a random baseline) to select a langauge to add to the training of an MT model are evaluated (one based on linguists' clustering, one based on probabilities assigned through a language model of the target langauge) on pairs drawn from a set of three langauges, choosing new langauges from a set of 13.

#### Strengths/what I loved:
- The paper performs a very clearly motivated and well set-up experiment with appropriate analysis.
- I loved to see a focus on datasets for underresourced langauges and the paper's repeated focus on whether or not previous work was reproducible and how this paper tries to be as reproducible as possible (even if that is not possible with Bible data for legal reasons).
- I always like to see attention to detail and explanation as i the "Due to the multi-task nature..."-paragraph on page 7!
- The observation that adding weakly related languages *hurt* was very interesting to see, thank you for including it!

#### Criticism/weaknesses:
- While the overall setup (choose a language to add using either linguistic knowledge or automatic metrics) is solid and interesting, the setup and contributions promised on pg. 2 make it sound especially like you are proposing a way to infer linguistic relations between languages ("a method for aggregating languages based on [a number of features]").
- The related work section is rather unfocused: it might help to make clear at the beginning of each paragraph how all the work cited in that paragraph relates to the present paper (e.g., "There are a number of papers on translation of African langauges, covering English to Setswana (Abbott & Martinus (2018); Transformers on the Autshumato dataset (Groenewald & Fourie (2009) reaching BLEU 33.53), English to Afrikaans..." or something along these lines. The division between the first and the second paragraph isn't quite clear to me and I am not sure what the third paragraph is on: transfer in MT?
- Not only training, but unfortunately also validation is performed only on Bible data and only on a cluster of 3 languages. The first is an issue because while I understand that Bibles are the easiest data to come by and that for training one takes what one can get, any results that stay within this Bible domain are unlikely to transfer well to real-world settings, making me doubt the practical use of the paper as-is. The latter exacerbates the issue, because essentially as a paper motivated by poor MT performance or nonexistence on many African languages it fails to convince me that it is indeed solving that task and not one of Bible translation, and as a paper that talks about linguistic features in general, the extremely narrow focus on a cluster of literally only 3 languages seems unnecessarily restrictive if we are already settling for Bible data. (Note that I think it is fair to say that these limitations were necessary because individual experiments are too costly to run more than this very barebones proof-of-concept, but that claim should be made and substantiated.)
- Experiments are not tested for statistical significance or otherwise qualified through a sense of the variance that's inherent in these results. This is especially unfortunate for the "Random" baseline, which given the small set of candidates will produce rather unreliable estimates when only tested on 1 sample (please let me know if I misunderstood!)

#### Questions:
- Point 2 of the Motivation section claims that language barriers aid the spread of misinformation... that's not quite clear to me: shouldn't any barrier impede information flow and in fact easy machine translation help the spread? Unless it is correcting counter-information that you want to advocate for...
- Page 5 says you used an English tokenizer for all languages. Just to clarify: this is tokenizing before applying BPE, right? Did you try to go without tokenization at all, i.e., using BPE as a standalone tokenizer?
- What is the point of the last two paragraphs of section 5, i.e., talking about clustering? From my understanding you don't use that for your experiments, as there you only choose the 1 nearest neighbor---and if you did want to build larger clusters, the greedy approach sketched in here would be a needlessly restricted choice: there are a number of better agglomerative clustering algorithms out there!
- The paper does cite the Masakhane project preprint from March 2020 (though with a broken arXiv identifier in the references!), but I would be very curious to see how it relates to the newer October 2020 preprint that contains a lot more content. (Of course, I know that this paper is far too recent to expect or even ask you to refer to it in the preprint! I would just personally be curious and think it might be a nice thing to add for a camera-ready.)


#### Typos and other small things:
- Probably due to my unfamiliarity, I wasn't 100% clear on what you meant by ``vernacular'' throughout, it might be nice to give a quick definition.
- Some small grammar errors throughout, for example "as follow" instead of "as follows" in a few places.
- Citations should be in parentheses when they are not constituents in a sentence and bare when they are (as in the beginning of the second paragraph on page 3). The citation of Lample and Conneau on page 2 somehow ended up with double parentheses...
- I would have liked to see references for section 2, but I suppose that is not really the scope or point of the papers. I also guess that the third point implies that colonial languages are learned instead of vernaculars? That might get expanded a little, too.
- The JW300 dataset should be properly cited (Agić and Vulić, 2019: https://www.aclweb.org/anthology/P19-1310/ ).
- The description of MLM on page 4 says "we keep them unchanged 10%"---this sounds like this is a deviation you are introducing.
- The entire first half of page 6 could be condensed into a much smaller and more concise paragraph, namely a sentence like the one that then starts with "At a paragraph level", or to more clearly separate motivation from actual definition (which is nice and simple)
- I would separate the top and bottom half of Table 1 more clearly, or better yet, make them separate tables altogether, given how they don't even both have the same set of columns.
- I fear the example given in the first paragraph of page 7 is a poor example: I see no reason a priori why learning a different unrelated language should *hurt* other languages I know. Then again, as a non-native speaker of English, my own native langauge certainly has taken the back seat in my head... either way it might be good to elaborate a little on this idea or just leave it out.

Looking forward to discuss!

---

> ### Author Response · Authors · 2020-11-21
> **A few attempts to answer the questions.**
>
> Dear Reviewer 4 , thank you for your positive feedback and constructive suggestions. We are in the process of incorporating them into the paper.
>
> * In relation to point 2 of the motivation section, we take an example relating to the election period. During this period, the government sends people to explain to people in rural areas what the vote is going to be used for and how they should vote. But it is common for them to do this in a way that favours certain political parties (especially the one in power) instead of simply translating the law, for example. Let's imagine that the text says in English "... You must therefore choose one and only one candidate between Candidate A and Candidate B." (... vous devez donc choisir un et un seul candidat entre le candidat A et le candidat B) and the translator says in French (to favour party A): "... vous devez donc choisir le candidat A et non le candidat B" (You must therefore choose candidate A and not candidate B). On the other hand, the information that is broadcast on television or radio is in French or English. If the villagers had access to a tool allowing them to translate the legal texts (which are in French or English) into their mother tongue, this could be avoided.
>
> * About page 5 : it is indeed a tokenizer before applying the BPE. We have not tried to do without tokenisation, this will be done later. Thank you for the suggestion.
>
> * "point of the last two paragraphs of section 5": the method of creation that we propose is only one agglomerative method among many others. We did not find it necessary to use very elaborate methods since the cluster was of size 3 (3 languages since the historical cluster 1 also had three languages).
>
> * For the definition of the word "vernacular", we found the one given by wikipedia [1] complete enough to meet the need, we will add it to the paper.
>
> * "...I see no reason a priori why learning a different unrelated language should hurt other languages I know..." : In our case, language learning takes place simultaneously, and we believe that learning several languages that are not similar (in the sense of the two similarities mentioned in the paper) simultaneously makes the learning task more difficult. we also believe that the gain in performance at the level of the historical clusters comes from a natural similarity in the Historical clusters. In Cameroon you will have difficulty, if you are not very good at speaking these languages, in differentiating between the languages of the same historical cluster (both alphabetically and orally). On the other hand, you can very easily, even if you are not familiar with these languages, make the difference between two languages of two different clusters
>
>
> [1] https://en.wikipedia.org/wiki/Vernacular

---

### Official Review · AnonReviewer3 · 2020-10-28
**Laudable effort! But little awareness of previous work**

**Rating:** 4
**Confidence:** 5

**Review:**

This paper considers translation between African languages. Overall, I think this is a great effort, I think it's great that the authors are tackling this important problem.

However, the field of multilingual machine translation is a very well-researched field, and it seems that the authors have developed their methodology largely independent of the literature in this field. In fact, there are already existing well-researched methods on many of the topics presented in this paper. To give just a few examples:

* *Pre-training for low-resource translation:* Liu, Yinhan, et al. "Multilingual denoising pre-training for neural machine translation." arXiv preprint arXiv:2001.08210 (2020).
* *Leveraging linguistic similarities:* Lin, Yu-Hsiang, et al. "Choosing transfer languages for cross-lingual learning." arXiv preprint arXiv:1905.12688 (2019).
* *Translation between low-resource language pairs:* Chen, Yun, et al. "A teacher-student framework for zero-resource neural machine translation." arXiv preprint arXiv:1705.00753 (2017).

I would suggest that the authors read these papers, and maybe other papers that cite them. Also, perhaps read papers on the ACL Anthology (https://www.aclweb.org/anthology/) from prominent conferences such as ACL, EMNLP, NAACL that contain the keywords "multilingual" and "translation" to get a better idea of the state of the art in the field. There are lots of methods that people have developed, and I think that they could be effectively applied to the very important problem at hand here!

---

> ### Author Response · Authors · 2020-11-22
> **state of the art in the African context**
>
> Dear Reviewer 3, thank you for your positive feedback and constructive suggestions.
>
> We are going to study these papers in depth and see how to apply them in our case. We have focused more on the state of the art in the African context in our paper, i.e. exploiting the multilingual aspect (and the associated similarity) encountered in the context of African languages to improve the performance of machine translation systems; something that has long been unexploited by existing research. From a replicability point of view, XLM was the best choice to conduct our experiments.

---

### Official Review · AnonReviewer1 · 2020-10-28
**Good work on an understudied area, but light on details for replication.**

**Rating:** 4
**Confidence:** 4

**Review:**

This paper describes three contributions: (1) a new multilingual parallel corpus that covers 1,477 languages, with text from the Bible -- for the region of interest in this paper (Cameroon) this means 22 languages instead of 18 with JW300; (2) a method for determining similarity of languages based on LM scores; (3) a series of experiments that evaluate the utility of adding a third language to a pair of interest based on either the similarity metric proposed by (2) or based on historical data, for use in a multilingual MT system. The authors show large improvements by adding a third language based on historical language similarity (always outperforms random) or their language-model-derived similarity (often outperforms random).

Working on MT for African languages is important, and I’d definitely like to see more of it. Unfortunately, I think this paper suffers substantially from insufficient replication details. Even if those details were present, I don’t think it is necessarily sufficiently novel for ICLR.

Regarding replication, the paper relies on three technologies: language modeling for the language similarity calculation, CLM and MLM pretraining, and multilingual MT. Crucial details are missing for all three of these. For the language modeling and for CLM and MLM pretraining, it isn’t clear what the monolingual data source is. Without being specified, it’s implied that it’s just one side of the parallel corpora - but why would pretraining help in that setting if it’s not adding data? In general, the data should be discussed with more precision. It would be really useful to talk about sizes of datasets (if the different languages have differently sized datasets).
For multilingual MT, it isn’t clear what flavor of multilingual MT is being used: is it a tagged model? Where are the tags (source versus target versus embedding)? I understand that the authors have provided code, but the paper should stand alone, and I wasn’t able to easily answer these questions after spending 15 minutes with the code.

Regarding novelty, I think the paper would need to flesh out either contributions (2) or (3) to really find a home at a venue like ICLR. (2) is an interesting idea, but right now we have one extrinsic test of the idea - I think a fully fleshed out version would need to compare to baselines and also provide intrinsic evaluations that compare to other “ground truth” definitions of similarity, such as the historical definition used in this paper. (3) would require looking into why we’re seeing such huge jumps just from adding another language. Are there disparities in dataset sizes that aren’t mentioned in the paper? If adding another language at most doubles the amount of parallel data, I don’t see why that should reflect a boost from 12.6 to 28.8 BLEU. There might be something really interesting going on here, but it would need to be explored much more carefully.

Smaller points:

When citing Eberhard, I would include “M.” with the first name, not the last name.

The proposed similarity metric is based on an equation normalizing a perplexity difference according to a base perplexity. But I don’t think this can be treated as a similarity, as lower is better in perplexity. So if I have language A that receives a perplexity of 10 according to LM0, and another language B that receives a perplexity of 3 (and thus much higher average probability per token), and LM0 assigns its own language a perplexity of 2, then we’ll get scores of (10-2)/2 = 4 for A and (3-2)/2 = 0.5 for B, making A look more similar. I think you probably intended to use average probability or average log probability here, or have the equation calculate a difference.

---

> ### Author Response · Authors · 2020-11-21
> **Some details about the training, the proposed similarity metrics and the results.**
>
> Dear Reviewer 1, thank you for your positive feedback.
> * Concerning pre-training, we only did MLM and TLM (without CLM), as mentioned in section 6.
>     - The MLM pre-training was used as the languages did not have the same number of sentences: for example for the Bafia-Bulu pair, we have 7950 parallel sentences (representing the number of sentences of the Bafia language) but 28895 sentences for Bulu (i.e. 28895 - 7950 = 20945 non-parallel sentences).
>     - The TLM is naturally motivated by XLM: encourage the model to align source and target sentence representations (especially by taking advantage of the target context if the source context is not sufficient to infer hidden source words).
>     - We have added the dataset sizes in the paper.
>
> * The similarity method we propose here is mainly motivated by the fact that the resources needed to apply other methods such as those mentioned in the paper are not available.
> 	- M. Eberhard et al (2020) has been easy to replicate in that it directly provides groups of similar languages: we have just made an intersection between these groups and the languages in our dataset.
> 	- On the other hand, a method like (Swadesh, 1952) requires parallelism of data at the word level, which we did not have and could not find: we will work on this later.
> * The difference between Random and Pair is not due to a disparity in data.
> 	- (a) They come from the same domain, the biblical domain.
> 	- (b) With regard to the size of the dataset.
> 		- For Bafia-Bulu, the three languages added are Du_n, Ewondo and Mofa for Random (R), Historical (H) and LMS (LANGUAGE MODEL BASED APPROACH FOR LINGUSTIC SIMILARITIES) respectively: this makes three clusters of almost the same size in terms of data, since only Bulu (present in all three clusters) among all these languages has 28895 sentences, the others have an average of 7939 (average of the sentences of the languages present only in the new testament).
> 		- Same thing for the Bulu-Ewondo pair with Dii, Bafia and Mofa added as languages for R, H and LMS respectively.
> 		- The pair which is an exception to this rule is Bafia-Ewondo, for which the languages added are respectively Guidar, Bulu and Bulu for R, H and LMS : H and LMS thus have an advantage over R with Bulu, but the gain is quite as good as for the other clusters.
> 	- (c) "There might be something really interesting going on here ..." : We believe that this comes from a natural similarity in the Historical clusters. In Cameroon you will have difficulty, if you are not very good at speaking these languages, in differentiating between the languages of the same historical cluster (both alphabetically and orally). On the other hand, you can very easily, even if you are not familiar with these languages, make the difference between two languages of two different clusters. We believe that this is one of the causes of this gain in performance.
>
> * The proposed similarity metric should be interpreted more as a dissimilarity, hence the equation normalizing a perplexity difference according to a base perplexity, to allow it to be zero on the base language and the base corpus. The word "similarity" here is only an abuse of language, to signify the fact that obenuous languages are similar in the sense of the proposed metric.

---

### Decision · Program_Chairs · 2021-01-07
**Final Decision**

**Decision:**

Reject

**Comment:**

This paper introduces a new multilingual parallel Bible dataset for African languages, a new method for determining similarities between languages, and a collection of experiments to evaluate methods for choosing an additional language based on (a) similarity and (b) language history to include in a multilingual MT system. Results show that strategic inclusion of an additional language can substantially improve BLEU. Reviewers universally agree that progress on MT for African languages is a very important goal. However, reviewers pointed to several major concerns with the current draft: (1) lack of sufficient detail for replicating experiments, (2) missing analysis to interpret why experimental gains are so large, and (3) missing discussion and comparison with already existing methods in multilingual MT (e.g. multilingual training for low-resource languages). I agree with reviewers that the paper is not ready for acceptance in its current form, but encourage re-submission, possibly at an NLP conference.